# Insertion/Deletion (InDel) Variants within the Sheep Fat-Deposition-Related *PDGFD* Gene Strongly Affect Morphological Traits

**DOI:** 10.3390/ani13091485

**Published:** 2023-04-27

**Authors:** Yunyun Luo, Mengyang Zhang, Zhengang Guo, Dwi Wijayanti, Hongwei Xu, Fugui Jiang, Xianyong Lan

**Affiliations:** 1Key Laboratory of Animal Genetics, Breeding and Reproduction of Shaanxi Province, College of Animal Science and Technology, Northwest A&F University, Yangling 712100, China; 2Bijie Animal Husbandry and Veterinary Science Research Institute, Bijie 551700, China; 3College of Life Science and Engineering, Northwest Minzu University, Lanzhou 730030, China; 4Shandong Key Lab of Animal Disease Control and Breeding, Institute of Animal Science and Veterinary Medicine, Shandong Academy of Agricultural Sciences, Jinan 250100, China

**Keywords:** sheep, *PDGFD* gene, insertion/deletion (InDel), morphological traits, association

## Abstract

**Simple Summary:**

Among the most important factors in the production of livestock are morphological traits, such as body weight, body length, chest depth, chest width, cannon circumference, and the body index. Platelet-derived growth factor D (*PDGFD*) is a candidate gene that has the potential to affect fat deposition and body size in sheep. In this study, we discovered two intronic InDels (13 bp deletion, 14 bp insertion) with the potential to significantly affect the morphological traits of three different indigenous Chinese sheep breeds. These InDels can be used as DNA markers for sheep marker-assisted selection (MAS) breeding.

**Abstract:**

Platelet-derived growth factor D (*PDGFD*) is a member of the PDGF gene family, and it plays an important role in the regulation of adipocyte development in mammals. Furthermore, genome-wide association studies (GWAS) have previously identified it as a candidate gene associated with fleece fiber variation, body size, and the fat-tail phenotype in domestic Chinese sheep. In this study, a total of 1919 indigenous Chinese sheep were genotyped to examine the association between nucleotide sequence variations in *PDGFD* and body morphology. Our results detected both a 14 bp insertion in intron 2 and a 13 bp deletion in intron 4 of *PDGFD*. Moreover, these two InDel loci had low to moderate polymorphism. Notably, the 13 bp deletion mutation of *PDGFD* was found to significantly affect sheep body size. Yearling rams in the Luxi black-headed sheep (LXBH) containing a heterozygous genotype (insertion/deletion, ID) were found to have larger body length, chest depth, and body weight than those with wild genotypes. Furthermore, adult ewes in the Guiqian semi-fine wool sheep (GSFW) containing a homozygous mutation (deletion/deletion, DD) were found to have smaller chest width than their peers. Moreover, yearling ewes in this group with the same homozygous mutation were found to have lower body weight, chest width, and cannon circumference compared to those of other individuals. This study demonstrates that *PDGFD* InDel polymorphisms have the potential to be effective molecular markers to improve morphological traits in domestic Chinese sheep.

## 1. Introduction

China has many extensive and genetically diverse types of sheep that are well adapted to local agricultural and climate conditions, being resistant to common diseases [1]. Lanzhou fat-tailed sheep (LFT) have long tails and deposit the majority of their adipose tissue within them, thus allowing them to adapt well to harsh climates and environmental conditions [2,3]. Luxi black-headed sheep (LXBH) have short tails with plenty of fat, as well as a high fertility rate and excellent meat production capacity [4,5]. Furthermore, Guiqian semi-fine wool sheep (GSFW) are selected for their dual purpose of high mutton and wool production [6]. Increasing sheep productivity and improving meat quality are consistent priorities necessary for the continuous development of the Chinese sheep breeding industry. Livestock growth and carcass traits belong to the medium heritability range (0.20–0.35); thus, the accuracy of sheep selection and breeding can be improved by gene marker-assisted and genomic selection of candidate genes related to the above traits [7,8].

Previous selective sweep analyses and genome-wide association studies (GWAS) identified several key candidate genes functionally associated with growth, body size, and tail morphology in sheep, such as platelet derived growth factor D (*PDGFD*), bone morphogenetic protein 2 (*BMP2*), thyroid-stimulating hormone receptor (*TSHR*), xylulokinase (*XYLB*), and fibroblast growth factor 7 (*FGF7*) [9,10]. In recent years, the *PDGFD* gene was discovered to belong to the *PDGF* family, which encodes for a mitogenic factor used by mesenchymal cells and is located in the blood serum [11]. Moreover, *PDGFD* contains a tissue-specific expression pattern. It is expressed more abundantly in the pancreas, pituitary gland, ovaries, and adipose tissue than in other tissues in humans [12]. Studies have shown that *PDGFD* is involved in various intracellular signaling pathways, (PI3K/Akt, MAPK, mTOR, and Notch signaling) and regulates IGF1R, VEGF, and Snail, among other proteins. By affecting these pathways, *PDGFD* has the potential to regulate multiple-organ fibrosis, atherosclerosis, tissue repair, and cancer in humans, as well as fleece fiber diameter in domestic sheep [13,14,15,16].

Furthermore, *PDGFD* is highly conserved amongst different species [17]. In a recent study, one cattle nonreference sequence (NRS) variant overlapping with the intergenic region between the protein-coding genes *PDGFD* and *DYNC2H1* (dynein cytoplasmic 2 heavy chain 1) was identified by multiple de novo assemblies, which may have biological significance [18]. *PDGFD* was also shown to affect intermuscular fat deposition in Dianzhong cattle [19]. Furthermore, *PDGFD* plays a critical role with regard to adipocyte proliferation and differentiation, and it is directly and indirectly involved in fat metabolism as it relates to the tails of sheep [10]. There are two single-nucleotide polymorphisms (SNPs) within the *PDGFD* gene that have been shown to significantly affect the size of sheep tails [20]. These studies suggest that *PDGFD* is closely related to sheep growth and development. However, it is notable that the functional role of *PDGFD* polymorphisms in relation to morphological traits of sheep is still poorly understood. Therefore, we aimed to characterize InDel variations of *PDGFD* in three different Chinese sheep breeds to validate the association between InDel polymorphisms and morphological traits, as well as provide fundamental reference data for use of *PDGFD* in domestic sheep breeding.

## 2. Materials and Methods

### 2.1. Animals, Phenotypic Data, and DNA Extraction

A total of 1919 sheep of three breeds were chosen for this study: Guiqian semi-fine wool sheep (GSFW, n = 1243, Bijie, China), Luxi black-headed sheep (LXBH, n = 618, Liaocheng, China), and Lanzhou fat-tailed sheep (LFT, n = 65, Lanzhou, China). Of the 618 LXBH sheep, 37.2% (n = 230) were lambs (≤3 months), 44.3% (n = 274) were yearlings (4–18 months old), and 18.5% (n = 114) were adults (>18 months). Among the GSFW group, 477 animals already had body size records (yearling rams, n = 143; yearling ewes, n = 155; adult ewes, n = 179). Within each sheep breed, healthy sheep individuals raised under similar feeding and management conditions were selected as tested sheep in this study. Morphological characteristics, i.e., body weight, body height, body length, chest circumference, chest depth, chest width, and cannon circumference, were recorded for each sheep in the study [21,22]. On the basis of prior established formulas, body size indices, such as body trunk index (the ratio of chest circumference to body length), body length index (the ratio of body length to body height), chest width index (the ratio of chest width to chest depth), chest circumference index (the ratio of chest circumference to body height), cannon circumference index (the ratio of cannon circumference to body height), and limb length index (the ratio of the difference between body height and chest depth to body height), were also calculated [23,24]. Furthermore, genomic DNA was extracted from ear tissues using standard phenol/chloroform extraction procedures as described previously and diluted to 20 ng/μL with ddH_2_O after determination of concentration [25]. To study genetic differences in the *PDGFD* gene, 30 DNA samples from each sheep breed were chosen at random and placed into three different DNA pools [26,27,28].

### 2.2. InDel Detection and Genotyping

The Ensembl database was used to find six possible InDel sites within sheep *PDGFD* gene. NCBI Primer-Blast was subsequently used to design six pairs of primers (Table 1) based on the *PDGFD* reference sequence for sheep (NC_056068.1). These primers were synthesized via Sangon Biotech (Xi’an, Shaanxi, China). PCR was performed using a 13 μL reaction volume and Touchdown program described previously [25,26]. PCR products were then directly examined via electrophoresis with a 3% agarose gel for confirmation.

### 2.3. Statistical Analyses

Genotype frequency, allele frequency, and population indices (heterozygosity, He; the number of effective alleles, Ne; polymorphism information content, PIC) were all calculated using Microsoft Excel (2019). Population indices were calculated following Nei’s methods [29]. In order to check whether the polymorphisms deviated from the Hardy–Weinberg equilibrium (HWE), the chi-square test (χ^2^) was carried out. Differences in allele frequencies, as well as the genotypic distribution, for each sheep population were assessed using the χ^2^ test or Fisher’s exact test. If all expected counts were greater than 5, the Pearson chi-square test was used. If there was a certain expected count <5, the Fisher exact test was used. The SHEsis online platform (http://analysis.bio-x.cn, accessed on 4 October 2022) was used for linkage disequilibrium (LD) analysis. The association between different genotypes of InDels and morphological traits of each sheep breed was evaluated using the Student’s *t*-test (only two genotypes) or one-way ANOVA (three genotypes) via SPSS 25 software. The statistical model used was as follows: Y_ijk_ = µ + G_i_ + S_j_ + A_k_ + e_ijk_, in which Y_ijk_ represents the phenotypic value for morphological traits, μ represents the population mean, G_i_ represents the fixed *PDGFD* genotype effect for each group of sheep, S_j_ represents the fixed effect of gender, A_k_ represents the fixed effect of the age, and e_ijk_ represents the random error.

## 3. Results

### 3.1. InDel Genotyping and Sequencing

On the basis of the combined results of gel electrophoresis, as well as sequencing profiles, two noncoding InDel mutation sites within *PDGFD* gene were detected: a 14 bp insertion (rs590816164) in intron 2, and a 13 bp deletion in intron 4 (rs1092650847) (Figure 1). Genotyping analysis showed that there were two genotypes at the P4-ins-14bp site (insertion/deletion, ID, 312 bp/298 bp; deletion/deletion, DD, 298 bp) (Figure 2), and three genotypes at P5-del-13bp site (insertion/insertion, II, 185 bp; ID, 185 bp/172 bp; DD, 172 bp) (Figure 3). The P4-ins-14bp locus was polymorphic for both LFT and GSFW sheep, whereas the P5-del-13bp showed genetic polymorphisms amongst all three sheep breeds.

### 3.2. Genetic Parameters and Linkage Disequilibrium Analysis

Genotypes, allele frequencies, and population genetics analyses are shown in Table 2. Amongst all sheep breeds, the frequency of the wild genotype was greater than 0.6 at both InDel loci. Furthermore, wildtype frequency was found to be greater than the frequency for both homozygous mutants and heterozygotes. Population genetic analyses indicated that the P4-ins-14bp locus demonstrated low levels of polymorphism (PIC < 0.25) amongst all tested sheep breeds. Furthermore, the P5-del-13bp demonstrated low levels of polymorphism (PIC < 0.25) for Luxi black-headed sheep and Lanzhou fat-tailed sheep, and moderate levels of polymorphism (0.25 < PIC < 0.5) for Guiqian semi-fine wool sheep. Furthermore, in GSFW populations, the P4-ins-14bp locus was found to deviate from the Hardy–Weinberg equilibrium (*p* < 0.05). No significant deviations from HWE were identified for the P5-del-13bp locus in all tested sheep breeds (*p* > 0.05).

Chi-square test results pertaining to the allelic frequency distribution of both InDel mutation sites amongst all three sheep breeds showed that the allele and genotype frequencies of the P4-ins-14bp locus were significantly different amongst all three breeds (*p* < 0.05) (Table 3). Furthermore, at the P5-del-13bp locus, the frequencies for both genotype and allele were considered statistically different amongst all three breeds (Table 4). According to linkage disequilibrium analysis (Figure 4), both P4-ins-14bp and P5-del-13bp loci showed a strong LD state within the LFT (D′ = 1, r^2^ = 0) population, and a weak linkage state in the GSFW (D′ = 0.060, r^2^ = 0.001) population.

### 3.3. Association Analysis of PDGFD InDel and Body Morphometric Traits

The association between genotype and phenotypic traits (seven body size traits and six body size indices) was analyzed. Significant deviations from HWE were identified for the P4-ins-14bp locus (*p* = 0.001142); thus, this site was not used for association analysis [30]. For the P5-del-13bp locus, significant differences were observed in chest depth, body length, and body weight amongst LXBH yearling rams (*p* < 0.05), and individuals with wild genotype had a smaller overall body size than individuals with a heterozygous genotype. Furthermore, a homozygous mutation (DD) at the P5-del-13bp locus showed a strong, negative influence on body weight, chest width, and cannon circumference in the GSFW sheep (*p* < 0.05). The body weight, chest width, and cannon circumference of yearling ewes with DD genotypes were significantly smaller than those of other individuals (*p* < 0.05). The chest width of adult ewes with a homozygous mutation was significantly smaller compared to other individuals (*p* < 0.05) (Table 5). Some traits did not differ significantly amongst individuals with different genotypes, and these results are not shown in the results table.

## 4. Discussion

The *PDGF* family includes proteins that are both proangiogenic and regulatory factors stimulating connective tissue growth [31]. *PDGF* comprises four subtypes: *PDGFA*, *PDGFB*, *PDGFC*, and *PDGFD* [32]. A prior study suggested that *PDGFC* is involved in adipose expansion [33]. The protein coded for by *PDGFD* mediates PDGF receptor chain tyrosine kinase (PDGFR-β) dimer formation and activation [34]. *PDGFD* is associated with angiogenesis required for tissue development and growth due to the inclusion of PDGF receptors on the surface of adipocytes and pre-adipocytes [35]. Furthermore, *PDGFD* has been shown to play a key role in the formation and development of pre-adipocytes, which can be seen in all species of mammals [27].

Adipose tissue is the main means of caloric storage in the body and plays a crucial role in the regulation of metabolism and body shape [36]. The large amount of tail fat in fat-tailed sheep was shown to have a significant influence on fat deposition in other parts of their bodies [37]. Thus, elucidating genetic variations of genes related to fat metabolism and deposition, as well as analyzing how these variations associate with morphological traits may provide more information about useful molecular markers to genetically improve sheep morphology [38]. For example, the gene encoding for cyclic AMP response element-binding protein 1 (*CREB1*) regulates fat metabolism in sheep adipose tissue, with one mutation within its first intron region strongly associated with differences in body measurement traits [39,40]. *PDGFD*, which is located on chromosome 15, has also been identified as a candidate gene for the phenotypic traits of sheep (i.e., fat tail and body size) [5,10].

Related studies have shown that genes affecting animal reproductive traits may also affect animal morphological and growth-related traits. Recently, Kang et al. discovered that a 7 bp InDel variation in intron 8 of lysine demethylase 3B (*KDM3B*) not only affected the litter size of Australian white sheep, but was also significantly associated with the body size traits of Lanzhou fat-tailed sheep and Luxi black-headed sheep [41,42]. Previously, Su et al. discovered that an 18 bp deletion in intron 2 of *PDGFD* was associated with litter size for Australian white sheep [25]. Data from the Ruminant Genome and NCBI databases showed that *PDGFD* is highly expressed in the reproductive system (i.e., ovary) [25]. These results imply that *PDGFD* may control multiple traits simultaneously (i.e., gene pleiotropism).

With these studies in mind, *PDGFD* was considered as a potential candidate gene for our study. In this study, we detected two InDel mutations, P4-ins-14bp and P5-del-13bp, both of which were found to be in Hardy–Weinberg equilibrium (*p* > 0.05) for Lanzhou fat-tailed sheep and Luxi black-headed sheep. These results indicate that no large-scale migration or mutations occurred in the tested population. It is worth noting that the P4-ins-14bp InDel existed in both Lanzhou fat-tailed sheep (2.59%) and Guiqian semi-fine wool sheep (8.31%) populations with relatively low frequency; however, Luxi black-headed sheep populations did not have this InDel. This may be due to different breeding requirements for those sheep varieties or the relatively conservative nucleotide sequence of *PDGFD*, as its genetic variation has been shown to be species-specific and vary on the basis of population size with regard to analysis and testing [43]. The P5-del-13bp locus showed moderate polymorphism in Guiqian semi-fine wool sheep, indicating that this population has rich genetic diversity.

Furthermore, both P4-ins-14bp and P5-del-13bp loci were located in the intronic regions of *PDGFD*. Studies have shown that intronic mutations may function to regulate transcriptional activity by preventing or increasing transcription factor binding, thus influencing gene expression via splicing, which leads to phenotypic variation [44,45,46,47]. Wang et al. found an intron mutation at the seventh intron of human transcription factor hepatocyte nuclear factor 1A (*HNF-1A*) that caused abnormal mRNA splicing and impaired its activity as a transcription factor [48]. Li et al. identified nucleotide mutations within a cis-regulatory element in the bone morphogenetic protein receptor type-1B (*BMPR1B*) intron 1, which could control pig prolificacy via the cis-regulation of *BMPR1B* expression [49]. Therefore, we further investigated the effect of these two intronic mutations on the phenotypic variation of tested sheep. A strong genotype–phenotype association between the P5-del-13bp polymorphism and morphological traits of local sheep was observed. For Guiqian semi-fine wool sheep, the P5-del-13bp locus was found to negatively influence body weight, which is associated with negative responses in fleece production [50]. In the future, further studies are needed to explore the mechanism of phenotypic change caused by noncoding genetic variants. Overall, our findings indicate that *PDGFD* is a critical gene related to morphological traits in sheep.

## 5. Conclusions

In this study, a 14 bp insertion in intron 2 and a 13 bp deletion in intron 4 within the *PDGFD* gene were detected in multiple breeds of sheep. Additionally, the 13 bp noncoding genetic variation had a significant effect on morphological traits (body weight, body length, chest depth, chest width, cannon circumference, etc.) of Luxi black-headed sheep and Guiqian semi-fine wool sheep, indicating the possibility of using these InDel mutation sites as DNA markers to increase the growth and development of indigenous Chinese sheep. However, the concrete mechanisms via which *PDGFD* gene and the 13 bp deletion affect morphological traits are still indistinct.

## Figures and Tables

**Figure 1 animals-13-01485-f001:**
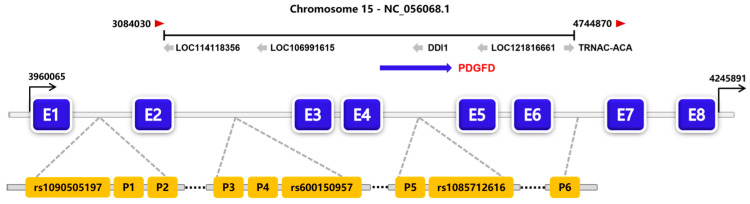
Position of the InDel variations in the genomic structure of the ovine *PDGFD* gene. Note: E1–E8: Exon 1–Exon 8; rs1090505197, rs600150957, and rs1085712616: InDel variations within ovine *PDGFD* gene in the study of Su et al. [25]; P1–P6: six possible InDel sites within ovine *PDGFD* gene in this study.

**Figure 2 animals-13-01485-f002:**
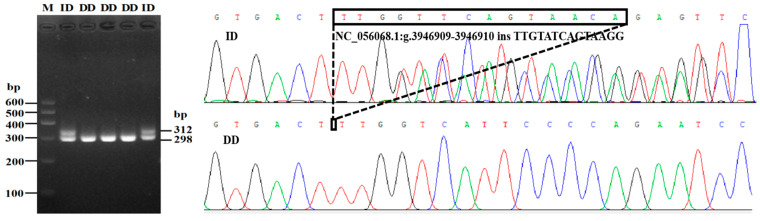
Agarose gel electrophoresis and sequence chromatograms for the P4-ins-14bp InDel locus.

**Figure 3 animals-13-01485-f003:**
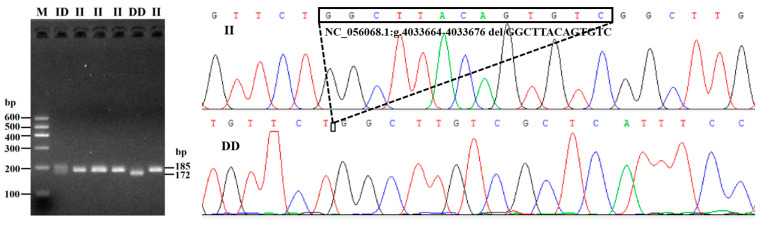
Agarose gel electrophoresis and sequence chromatograms for the P5-del-13bp InDel locus.

**Figure 4 animals-13-01485-f004:**
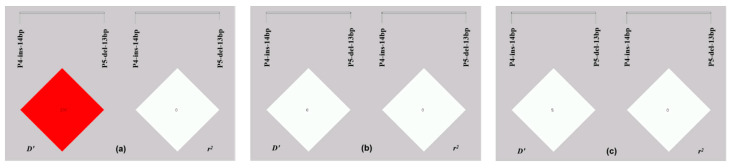
Linkage disequilibrium analysis of both P4-ins-14bp and P5-del-13bp loci in sheep *PDGFD* gene: (**a**) Lanzhou fat-tailed sheep; (**b**) Luxi black-headed sheep; (**c**) Guiqian semi-fine wool sheep.

**Table 1 animals-13-01485-t001:** Primers of the sheep *PDGFD* gene for genotyping.

Variant ID	Names	Primer Sequences (5′ to 3′)	Product Sizes (bp)	Location
rs1093585007	P1-del-39bp	F: TCAGAGGCTTGACTGAGGTATGA R: CAACATTCCACTCCGCAGTTT	378/339	Intron 1
rs1094063906	P2-del-18bp	F: GAAAGTCTGAAGGTGCCA R: CAGAGCTGATGCTGGAAT	150/132	Intron 1
rs604168702	P3-del-16bp	F: CACATCACTACTCCTTCACACAAC R: GCCTCATTCAAGAGTGGTCAGT	195/179	Intron 2
rs590816164	P4-ins-14bp	F: ACCCAGATCTTGGCCATTTTCTAT R: AGCCATGCTTTATTTCCAAAGTGG	312/298	Intron 2
rs1092650847	P5-del-13bp	F: TCGTTGAAGAAGGCAGTG R: GTGGAGGTGAATAAGTAAGTGA	185/172	Intron 4
rs594600476	P6-del-17bp	F: CAGACTCTGGCTCTTCTACTGA R: TGTAAAGCAACGGGACGT	291/274	Intron 6

**Table 2 animals-13-01485-t002:** Genetic parameters of InDels in three different sheep populations.

Loci	Breeds	Sample Sizes	Genotype and Allele Frequencies					He	Ne	PIC	HWE *p*-Value
			II	ID	DD	I	D				
P4-ins-14bp	LFT	65	0	0.062	0.938	0.031	0.969	0.060	1.063	0.058	*p* > 0.05
	LXBH	618	0	0	1.000	0	1.000	0	1.000	0	*p* > 0.05
	GSFW	1243	0	0.169	0.831	0.084	0.916	0.155	1.183	0.143	*p* < 0.05
P5-del-13bp	LFT	65	0.954	0.046	0	0.977	0.023	0.045	1.047	0.044	*p* > 0.05
	LXBH	618	0.835	0.160	0.005	0.915	0.085	0.155	1.184	0.143	*p* > 0.05
	GSFW	1243	0.635	0.322	0.043	0.796	0.204	0.325	1.481	0.272	*p* > 0.05

Note: LFT, Lanzhou fat-tailed sheep; LXBH, Luxi black-headed sheep; GSFW, Guiqian semi-fine wool sheep; II, insertion/insertion; ID, insertion/deletion; DD, deletion/deletion; He, heterozygosity; Ne, the effective allele numbers; PIC, polymorphism information content; HWE, Hardy–Weinberg equilibrium.

**Table 3 animals-13-01485-t003:** Chi-square testing of genotype frequency (below diagonal) and allele frequency (above diagonal) of the P4-ins-14bp locus in different sheep breeds.

Breeds	LFT	LXBH	GSFW
LFT	-	7.9 × 10^−5^ **	0.031 *
LXBH	7.5 × 10^−5^ **	-	9.11 × 10^−39^ **
GSFW	0.024 *	2.73 × 10^−40^ **	-

Note: * *p* < 0.05, ** *p* < 0.01.

**Table 4 animals-13-01485-t004:** Chi-square testing of genotype frequency (below diagonal) and allele frequency (above diagonal) of the P5-del-13bp locus in different sheep breeds.

Breeds	LFT	LXBH	GSFW
LFT	-	0.009 **	2.45 × 10^−9^ **
LXBH	0.035 *	-	9.41 × 10^−20^ **
GSFW	4.48 × 10^−8^ **	8.46 × 10^−21^ **	-

Note: * *p* < 0.05, ** *p* < 0.01.

**Table 5 animals-13-01485-t005:** The genetic and phenotypic association of the P5-del-13bp locus of *PDGFD* with morphological traits in sheep.

Traits	Observed Genotypes (LSM ^a^ ± SE)			*p*-Value
	II (n)	ID (n)	DD (n)	
	LXBH yearling rams (77)			
Body weight (kg)	54.10 ^b^ ± 2.32 (60)	64.18 ^a^ ± 3.84 (17)	-	0.040
Body length (cm)	70.78 ^b^ ± 0.87 (60)	75.82 ^a^ ± 1.84 (17)	-	0.010
Chest depth (cm)	27.28 ^B^ ± 0.58 (60)	30.76 ^A^ ± 0.98 (17)	-	0.005
	GSFW yearling ewes (155)			
Body weight (kg)	40.11 ^a^ ± 0.55 (103)	38.08 ^b^ ± 0.76 (48)	35.35 ^c^ ± 1.21 (4)	0.036
Chest width (cm)	26.41 ^A^ ± 0.24 (103)	25.14 ^B^ ± 0.26 (48)	25.00 ^C^ ± 0.71 (4)	0.005
Cannon circumference (cm)	9.14 ^A^ ± 0.34 (103)	9.23 ^A^ ± 0.06 (48)	9.00 ^B^ ± 0.00 (4)	3.53 × 10^−4^
Chest width index (%)	77.21 ^A^ ± 0.62 (103)	73.64 ^B^ ± 0.60 (48)	74.24 ^AB^ ± 3.15 (4)	4.98 × 10^−4^
	GSFW adult ewes (179)			
Chest width (cm)	28.68 ^A^ ± 0.22 (126)	28.50 ^A^ ± 0.35 (46)	27.14 ^B^ ± 0.14 (7)	8.67 × 10^−7^

Note: Values with different superscripts within the same line differ significantly at *p* < 0.01 (A, B, C) and *p* < 0.05 (a, b, c).

## Data Availability

Data are available upon request from corresponding author.

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
