# Peer review of "Insertion/Deletion (InDel) Variants within the Sheep Fat-Deposition-Related PDGFD Gene Strongly Affect Morphological Traits"

_animals, 2023, doi:10.3390/ani13091485_

Round 1

Author Response

Response: First of all, we wish to thank you for your recognition of the research significance of our manuscript. We have made a considerable effort to take into account the suggestions proposed by reviewers. The specific corrections in the paper and the responses to the reviewer’s comments are below:

Point 1: Title. I suggest replacing “fat-deposition related gene PDGFD” with “fat-deposition related PDGFD gene”.

Response 1: Thanks for your advice. The “fat-deposition related gene PDGFD” was corrected as “fat-deposition related PDGFD gene”.

Point 2: L27: It should be “intronic region”, not “intron region”.

Response 2: The corrections have been made in the revision.

Point 3: Introduction: The full name of each sheep breed should be followed by a corresponding abbreviation. Lanzhou fat-tailed sheep (LFT), Luxi black-headed sheep (LXBH); Guiqian semi-fine wool sheep (GSFW).

Response 3: Thanks for your reminding. According to your suggestion, we listed the corresponding abbreviations after the full name of each sheep breed.

Point 4: L68: “two SNPs at the PDGFD gene”, the preposition “at” does not seem appropriate here.

Response 4: Thanks for your careful suggestion. The “at” has been changed to “within”.

Point 5: L69: “18-bp nucleotide”. There are word duplications here. It should be “18-bp deletion” or “18 nucleotides deletion”.

Response 5: Many thanks for your careful comments. We have already modified “18-bp nucleotide deletion” to “18-bp deletion”.

Point 6: L90-92: Authors indicated DNA pools. Why are there 30 DNA samples in each pool?

Response 6: Referring to the published scientific research, 20-50 individuals were generally selected to construct a mixed DNA pool [1-4]. We selected the basic DNA pool unit consisting of 30 individual DNA samples.

  1. Huang, Y., Ma, Z., Wang, X., Zhao, L., Zhang, Y., Yang, X., Xu, D., Cheng, J., Li, X., Zeng, X., Zhao, Y., Li, W., Wang, J., Lin, C., Zhou, B., Liu, J., Zhai, R., Zhang, X. Identification of TRAPPC9 and BAIAP2 gene polymorphisms and their association with fat deposition-related traits in Hu sheep. Front Vet Sci 2022, 9, 928375.
  2. Li, Q.; Lu, Z.; Jin, M.; Fei, X.; Quan, K.; Liu, Y.; Ma, L.; Chu, M.; Wang, H.; Wei, C. Verification and Analysis of Sheep Tail Type-Associated PDGF-D Gene Polymorphisms. Animals 2020, 10(1), 89.
  3. Su, P.; Luo, Y.; Huang, Y.; Akhatayeva, Z.; Xin, D.; Guo, Z.; Pan, C.; Zhang, Q.; Xu, H.; Lan, X. Short variation of the sheep PDGFD gene is correlated with litter size. Gene 2022, 844, 146797.
  4. Wei, Z.; Wang, K.; Wu, H.; Wang, Z.; Pan, C.; Chen, H.; Lan, X. Detection of 15-bp deletion mutation within PLAG1 gene and its effects on growth traits in goats. Animals 2021, 11(7), 2064.

Point 7: L108-109: Please clarify the statistical analysis method used and when it was used. For example, when there were two genotypes, t-test might be used. when there are three genotypes, the ANOVA analysis might be applied.

Response 7: Thank you for your kind correction. Indeed, there are some small mistakes in the description of statistical analysis methods in our paper. In the revision, we have corrected this part. As follows:

The association between the different genotypes of InDels and growth traits was evaluated using the Student's unpaired t-test or one-way ANOVA in the SPSS 25 software. The statistical model was Yijkl = µ + Gi + Sj + Yk + eijkl, where Yijkl represents the phenotypic value of the growth traits; μ represents the population mean; Gi represents the fixed effect of PDGFD genotypes, Sj represents the fixed effect of gender, Yk represents the fixed effect of the age, and εijkl represents the random error.

Point 8: Figure 1. It is recommended to modify and beautify Figure 1 to make clearer exon and supplement the genomic positions of other published SNP and InDel variant sites.

Response 8: Thank you for your suggestion. We have already provided a new figure to make it clearer.

Point 9: Table 2. P4 locus showed only one genotype in LXBH sheep, so it was in Hardy-Weinberg equilibrium. There should be a HWE chi square p-value for P4 locus in LXBH sheep. It should not be empty. Please correct.

Response 9: Many thanks for your careful comments. We have changed “-“ to “P>0.05” in Table 2.

Point 10: Discussion. PDGFD affects litter size of sheep (Ref 20), and in this study, the authors found that PDGFD affects growth traits. How to explain that PDGFD regulates both reproduction and fat deposition? What’s the correlation between them? The authors are suggested to discuss this point in detail.

Response 10: Thank you for your question. We have added more discussion about the PDGFD gene function in the revised manuscript. As follows:

Related researches have shown that genes affecting animal reproductive traits may also affect the animal growth traits. Kang et al. found that a 7-bp InDel variation in intron 8 of Lysine demethylase 3B (KDM3B) gene not only had a significant effect on the litter size, live lamb rate of Australian white sheep, but also was significantly associated with the growth traits of Lanzhou fat-tailed sheep and Luxi black-headed sheep [1-2]. Our lab previously found that that an 18-bp deletion in intron 2 of PDGFD was associated with litter size in Australian white (AUW) sheep [3]. Data in the Ruminant Genome Database and NCBI database showed that PDGFD gene are highly expressed in the reproductive system, such as the ovary. These results imply that the PDGFD gene might control multiple traits simultaneously (i.e., the gene pleiotropism), that is PDGFD gene may be involved in the function of fat and ovary at the same time.

  1. Kang, Y., Bi, Y., Tang, Q., Xu, H., Lan, X., Zhang, Q., Pan, C. A 7-nt nucleotide sequence variant within the sheep KDM3B gene affects female reproduction traits. Anim Biotechnol 2022, 33(7), 1661-1667.
  2. Kang, Y., Zhu, Q., Meng, F., Xu, H., Guo, Z., Pan, C. Rapid detection of InDel within the KDM3B gene in five sheep breeds using the mathematical expectation (ME) method. Gene 2022, 834, 146598.
  3. Su, P.; Luo, Y.; Huang, Y.; Akhatayeva, Z.; Xin, D.; Guo, Z.; Pan, C.; Zhang, Q.; Xu, H.; Lan, X. Short variation of the sheep PDGFD gene is correlated with litter size. Gene 2022, 844, 146797.

    Point 11: Please check the references. The reference format should be standardized.

    For example: L340-341: ……Functional Screening of Candidate Causal Genes for Insulin Resistance in Human 340 Preadipocytes and Adipocytes……

    L344-345: ….. . Verification and Analysis of Sheep Tail Type- 344 Associated PDGF-D Gene Polymorphisms….

    Response 11: Thanks for your advice. We have modified and unified the format of the references according to the requirements of Animals journal.

Reviewer 2 Report

Line 20: MAS. When the first time it appears, you need to spell it all.

Line 25: correlation. No correlation analysis was performed in your study. It should be “association”.

Line 26: 13-bp deletion and a 14-bp insertion. How do you know this is a deletion or insertion mutation? Compared to what sequence?

Line 42: are should be is.

Line 55: “the” should be deleted.

Line 59: in humans should be moved to after “than in other tissues”.

Line 60: Cys/Tyr position?

Line 101: 2.3 Statistical analyses. You showed a completely wrong statistical method. The results of statistical analysis are not reliable.

Line 117: identified? I think “detected” is correct, because you did not discover the mutation.

Line 154: weak? LD state in the LFT (D′=1, r2=0), D’ is 1, this is strong or complete LD.

Line 163-166 and Table 5:  P4-ins-14bp locus with growth traits in yearling ewes of GSFW sheep. P4-ins-14bp locus deviated from the Hardy-Weinberg equilibrium (P<0.05) in GSFW populations. Thus, this site cannot be used for any analysis of genetic diversity and association.

On the whole:

Based on the wrong statistical analysis, the results of the statistical analysis are not reliable.

Author Response

Thanks for your time and efforts on the improvement of our manuscript. The manuscript has been greatly improved according to your suggestions. Here, we have listed the point-by-point responses to your detailed comments and suggestions.

Point 1: Line 20: MAS. When the first time it appears, you need to spell it all.

Response 1: Thank you for your careful comments. The full name has been given when the “MAS” listed for the first time. Now the sentence we have already modified to “which can be used as DNA markers in sheep marker-assisted selection (MAS) breeding.”

Point 2: Line 25: correlation. No correlation analysis was performed in your study. It should be “association”.

Response 2: Thank you for your correction. The correction has been made.

Point 3: Line 26: 13-bp deletion and a 14-bp insertion. How do you know this is a deletion or insertion mutation? Compared to what sequence?

Response 3: The two InDels loci were previously found on the Ensembl online database (http://asia.ensembl.org/), and their rs numbers were rs590816164, rs1092650847 respectively. The PCR products of mutant alleles were sequenced, and the sequencing results were compared with the wild-type genome sequence (NCBI reference sequence: NC_056068.1) to confirm the specific location of the mutant gene sequence in the genome and the number of mutant bases.

Point 4: Line 42: are should be is.

Response 4: “are” has been changed to “is”.

Point 5: Line 55: “the” should be deleted.

Response 5: “the” has been deleted.

Point 6: Line 59: in humans should be moved to after “than in other tissues”.

Response 6: The corrections have been made. Now the sentence we have already modified to “It is expressed more abundantly in the pancreas, pituitary gland, ovaries, and adipose tissue than that in other tissues in humans.”

Point 7: Line 60: Cys/Tyr position?

Response 7: In the revision, we rewrote this sentence as follow “amino acid change (cysteine to tyrosine) of the PDGFD protein have been linked to fat metabolism and adipogenesis”.

Point 8: Line 101: 2.3 Statistical analyses. You showed a completely wrong statistical method. The results of statistical analysis are not reliable.

Response 8: Thank you for your kind correction. Indeed, there are some small mistakes in the description of statistical analysis methods in our paper. In the revision, we have corrected this part. As follows:

The association between the different genotypes of InDels and growth traits was evaluated using the Student's unpaired t-test or one-way ANOVA in the SPSS 25 software. The statistical model was Yijkl = µ + Gi + Sj + Yk + eijkl, where Yijkl represents the phenotypic value of the growth traits; μ represents the population mean; Gi represents the fixed effect of PDGFD genotypes, Sj represents the gender effect, Yk represents the age age, and εijkl represents the random error.

Point 9: Line 117: identified? I think “detected” is correct, because you did not discover the mutation.

Response 9: Thanks for your advice. These polymorphisms are not novel since they were already described in Ensembl database. Thus, “detected” maybe was more appropriate word.

Point 10: Line 154: weak? LD state in the LFT (D′=1, r2=0), D’ is 1, this is strong or complete LD.

Response 10: Thank you for your correction. We have already rewrote this sentence.

According to the results of linkage disequilibrium analysis (Figure 4), the loci P4-ins-14bp and P5-del-13bp were in a strong LD state in the LFT population (D′=1, r2=0), and were in a weak linkage state in GSFW population (D′=0.060, r2=0.001).

Point 11: Line 163-166 and Table 5: P4-ins-14bp locus with growth traits in yearling ewes of GSFW sheep. P4-ins-14bp locus deviated from the Hardy-Weinberg equilibrium (P<0.05) in GSFW populations. Thus, this site cannot be used for any analysis of genetic diversity and association.

Response 11: Thanks for your advice. We have consulted relevant literatures, and found that mutation sites that do not meet Hardy-Weinberg equilibrium in the analysis of animal and plant association studies are generally not filtered out. However, there are some studies with stricter screening criteria that remove sites that seriously departed from Hardy-Weinberg equilibrium (P<0.01) [1]. P4-ins-14bp locus significant departure from HWE (P=0.001142), so this site really cannot be used for association analysis. In the revision, “Table 5. The genetic and phenotypic correlation of P4-ins-14bp locus with growth traits in yearling ewes of GSFW sheep” has been deleted.

  1. Zhou, Q.; Chen, Y.; Lu, S.; Liu, Y.; Xu, W.; Li, Y.; Wang. L.; Wang. N.; Yang. Y.; Chen. S. Development of a 50K SNP array for Japanese flounder and its application in genomic selection for disease resistance. Engineering 2021, 7(3), 406-411.

Point 12: On the whole: Based on the wrong statistical analysis, the results of the statistical analysis are not reliable.

Response 12: Thank you for your careful comments. We have comprehensively revised the statistical analysis part of this manuscript. Please see the response 8 for details. 

Reviewer 3 Report

This paper discusses how Insertion/deletion variants within sheep fat-deposition-related gene PDGFD strongly affect growth traits.

Some of the issues with the paper

1. The abstract needs to be rewritten completely. The sentences are badly constructed. Lines 25-26 talks about correlations between nucleotide sequence and body morphometric traits. It is important to note that correlations do not depicts causation and the use of that word in a paper that is trying to show causality is not appropriate.

2. Line 28- What does significant relevance mean? Please use the appropriate word.

3. Line 29 - There is no such thing as deletion or insertion loci It should be 13bp deletion within the gene sequence.

4. Line 30 - performed significantly lower body length is not a good sentence. It should be rephrased.

5. Lines 129 -131 - Genotypic and allelic frequencies and population genetic analyses should be moved to the materials and methods and expanded.

6. The mechanism of action of intronic variants should be explained.

7. Line 223 - 226 will need to be expanded

8. The conclusion should be more than one sentence.

9. The paper needs a lot more work.

Author Response

This paper discusses how Insertion/deletion variants within sheep fat-deposition-related gene PDGFD strongly affect growth traits.

Response: Thanks for your comments on our manuscript.

Some of the issues with the paper:

Point 1: The abstract needs to be rewritten completely. The sentences are badly constructed. Lines 25-26 talks about correlations between nucleotide sequence and body morphometric traits. It is important to note that correlations do not depicts causation and the use of that word in a paper that is trying to show causality is not appropriate.

Response 1: Thank you very much for your careful comments. We have already made some modification to our abstract. The specific version was as follows:

Platelet-derived growth factor D (PDGFD) is a member of the PDGF gene family, and plays an important role in the regulation of adipocyte development in mammals. Furthermore, genome-wide association studies (GWAS) have previously identified it as a candidate gene associated with fleece fiber variation, body size, and the fat-tail phenotype in domestic Chinese sheep. In this study, a total of 1919 indigenous Chinese sheep were genotyped to examine the association between nucleotide sequence variations in PDGFD and body morphology. Our results detected both a 14-bp insertion in intron 2 and a 13-bp deletion in intron 4 of PDGFD. Moreover, these two InDel loci had low to moderate polymorphism. Notably, the 13-bp deletion mutation of PDGFD was found to significantly affect sheep body size. Yearling rams in the Luxi black-headed sheep (LXBH) containing a heterozygous genotype (insertion/deletion, ID) were found to have larger body length, chest depth, and body weight than ones with wild genotypes. Furthermore, adult ewes in the Guiqian semi-fine wool sheep (GSFW) containing a homozygous mutation (deletion/deletion, DD) were found to have smaller chest widths than their peers. Moreover, yearling ewes in this group with the same homozygous mutation were found to have lower body weight, chest width and cannon circumference compared to those of other individuals. This study demonstrates that PDGFD InDel polymorphisms have the potential to be effective molecular markers to improve growth trait performance in domestic Chinese sheep.

Point 2: Line 28- What does significant relevance mean? Please use the appropriate word.

Response 2: The “relevance” has been corrected as “association”.

Point 3: Line 29 - There is no such thing as deletion or insertion loci It should be 13bp deletion within the gene sequence.

Response 3: We have already rewrote this sentence.

Point 4: Line 30 - performed significantly lower body length is not a good sentence. It should be rephrased.

Response 4: Thanks for your advice. We have already rewrote this sentence.

Yearling rams in the Luxi black-headed sheep (LXBH) containing a heterozygous genotype (insertion/deletion, ID) were found to have larger body length, chest depth, and body weight than ones with wild genotypes.

Point 5: Lines 129 -131 - Genotypic and allelic frequencies and population genetic analyses should be moved to the materials and methods and expanded.

Response 5: Thanks for your advice. We have already rewrote this part.

Genotype and allele frequency were calculated with the software Microsoft Excel (2019). Population indices including: heterozygosity (He), the number of effective al-leles (Ne), polymorphism information content (PIC) were calculated following Nei’s methods [1].

  1. Nei, M. Analysis of gene diversity in subdivided populations. Proc. Natl. Acad. Sci. USA. 1973, 70, 3321–3323.

Point 6: The mechanism of action of intronic variants should be explained.

Response 6: Thanks for your advice. This part has been improved in the revised manuscript.

Furthermore, both P4-ins-14bp and P5-del-13bp loci were located in the intronic regions of PDGFD. Studies have shown that intronic mutations may function to regulate transcriptional activity by preventing or increasing transcription factor binding, thus influencing gene expression via splicing, which leads to phenotypic variation [1-4]. Wang, et al., found an intron mutation at the seventh intron of human transcription factor hepatocyte nuclear factor 1A (HNF-1A) caused abnormal mRNA splicing, and impaired its activity as a transcription factor [5]. Li, et al., identified nucleotide mutations within a cis-regulatory element in the bone morphogenetic protein receptor type-1B (BMPR1B) intron 1, which could control pig prolificacy via the cis-regulation of BMPR1B expression [6]. Therefore, we further investigated the effect of these two intronic mutations on the phenotypic variation of tested sheep.

  1. Berulava, T.; Horsthemke, B. The obesity-associated SNPs in intron 1 of the FTO gene affect primary transcript levels. Eur J Hum Genet 2010, 18(9), 1054-1056.
  2. Cui, Y.; Yan, H.; Wang, K.; Xu, H.; Zhang, X.; Zhu, H.; Liu, J.; Qu, L.; Lan, X.; Pan, C. Insertion/Deletion within the KDM6A gene is significantly associated with litter size in goat. Front Genet 2018, 9, 91.
  3. Shaul O. How introns enhance gene expression. Int J Biochem Cell Biol 2017, 91(Pt B), 145-155.
  4. Yang, H.; Zhang, H.; Luan, Y.; Liu, T.; Yang, W.; Roberts, K. G.; Qian, M. X.; Zhang, B.; Yang, W.; Perez-Andreu, V.; Xu, J.; Iyyanki, S.; Kuang, D.; Stasiak, L. A.; Reshmi, S. C.; Gastier-Foster, J.; Smith, C.; Pui, C. H.; Evans, W. E.; Hunger, S. P.; Platanias, L. C.; Relling, M. V.; Mullighan, C. G.; Loh, M. L.; Yue, F.; Yang, J. J. Noncoding genetic variation in GATA3 increases acute lymphoblastic leukemia risk through local and global changes in chromatin conformation. Nat Genet 2022, 54(2), 170-179.
  5. Wang, M.; Shu, H.; Xie, J.; Huang, Y.; Wang, K.; Feng, R.; Yu, X.; Guan, J.; Feng, W.; Liu, M. An intron mutation of HNF1A causes abnormal splicing and impairs its activity as a transcription factor. Mol Cell Endocrinol 2022, 545, 111575.
  6. Li, W.; Zhang, M.; Li, Q.; Tang, H.; Zhang, L.; Wang, K.; Zhu, M.; Lu, Y.; Bao, H.; Zhang, Y.; Li, Q.; Wu, K.; Wu, C. Whole-genome resequencing reveals candidate mutations for pig prolificacy. Proc Biol Sci 2017, 284(1869), 20172437.

Point 7: Line 223 - 226 will need to be expanded.

Response 7: Thanks for your advice. In discussion part, we have added some explanation and elaboration about the mechanism of action of intronic variants. Please see the response 6 for details.

Point 8: The conclusion should be more than one sentence.

Response 8: Thank you for your useful suggestion, we have expanded the conclusion part following your advice.

In this study, a 14-bp insertion in intron 2 and a 13-bp deletion in intron 4 within the PDGFD gene were detected in multiple breeds of sheep. Additionally, the 13-bp noncoding genetic variation had a significant effect on growth traits (body weight, body length, chest depth, chest width, cannon circumference, etc.) of LXBH and GSFW sheep, indicating the possibility of using these InDel mutation sites as DNA markers to increase the growth and development of indigenous Chinese sheep. However, the concrete mechanisms by which PDGFD gene and the 13-bp deletion affect growth traits are still indistinct.

Point 9: The paper needs a lot more work.

Response 9: Deeply thanks for your careful comment.

In this study, we detected the polymorphisms within the sheep fat-deposition- related PDGFD gene in a total of 1919 Chinese indigenous sheep, and found a 13-bp deletion and a 14-bp insertion. Moreover, the 13-bp deletion mutation was found to significantly affect sheep body size. Therefore, InDel polymorphisms of PDGFD gene might be effective molecular markers to improve the performance in growth traits in Chinese native sheep.

Indeed, there are some defects in our research, which need to be further improved and supplemented.

For example, the sample size of different breeds of sheep was slightly different. Besides, the concrete mechanisms by which PDGFD gene and the 13-bp deletion affect growth traits are still indistinct. To comprehensive understanding the function roles of these noncoding genetic variations in the regulation phenotypic traits, further research and validation at the molecular and cellular level should be made.

Reviewer 4 Report

The authors investigated the influnence of Plateled-derived growth factor D (PDGFD) on fat deposition, and body size in sheep. A 13 bp deletion and a 14 bp insertion in the intron region were specifically investigated with the aim of their polymorphism using as a molecular marker to improve the performance in growth traits in Chinese native sheep. 

The authors performed a complete introduction, with all necessary details for knowing general aspects of the debated subject. 

Row 69 - Introduction: the sentence „We also found...” seems to reffer that the first author and his team, or members of his team found that association of the disscussed genetic marker. But, in reality, they reffer  to the fact that that information was also found in literature. This aspect has to be clarified by the authors. 

The population investigated includes enough animals for concludent results. Some details on work protocol of molecular biology were expected to be found, as well as data of genotypes consideration. These data of genotypes are presented in Results chapeter, but their reorganization in Material and Methods may be considered. 

An interpretation of „a” and „b” indexes used in Table 6 is better to be given at the bottom of this table. 

Author Response

The authors investigated the influnence of Plateled-derived growth factor D (PDGFD) on fat deposition, and body size in sheep. A 13 bp deletion and a 14 bp insertion in the intron region were specifically investigated with the aim of their polymorphism using as a molecular marker to improve the performance in growth traits in Chinese native sheep. The authors performed a complete introduction, with all necessary details for knowing general aspects of the debated subject.

Row 69 - Introduction: the sentence „We also found...” seems to refer that the first author and his team, or members of his team found that association of the discussed genetic marker. But, in reality, they refer to the fact that that information was also found in literature. This aspect has to be clarified by the authors.

Response: Thank you for your valuable advice. Su et al. [1] found that an 18-bp deletion (rs1090505197) in intron 2 of PDGFD gene was associated with litter size in Australian white sheep. Related researches have shown that genes affecting animal reproductive traits may also affect the animal growth traits [2-3]. Thus, we further study the association between PDGFD gene polymorphisms and sheep growth traits. Firstly, we detected the polymorphism of rs1090505197 locus in Lanzhou fat-tailed sheep and Luxi blackhead sheep. Unfortunately, this locus was not polymorphic in both sheep breeds. Therefore, we further retrieved other InDel loci of PDGFD gene from Ensembl database for research. Finally, a 13 bp deletion and a 14 bp insertion in the intron region were detected.

  1. Su, P.; Luo, Y.; Huang, Y.; Akhatayeva, Z.; Xin, D.; Guo, Z.; Pan, C.; Zhang, Q.; Xu, H.; Lan, X. Short variation of the sheep PDGFD gene is correlated with litter size. Gene 2022, 844, 146797.
  2. Kang, Y.; Bi, Y.; Tang, Q.; Xu, H.; Lan, X.; Zhang, Q.; Pan, C. A 7-nt nucleotide sequence variant within the sheep KDM3B gene affects female reproduction traits. Anim Biotechnol 2022, 33(7), 1661-1667.
  3. Kang, Y.; Zhu, Q.; Meng, F.; Xu, H.; Guo, Z.; Pan, C. Rapid detection of InDel within the KDM3B gene in five sheep breeds using the mathematical expectation (ME) method. Gene 2022, 834, 146598.

The population investigated includes enough animals for concludent results. Some details on work protocol of molecular biology were expected to be found, as well as data of genotypes consideration. These data of genotypes are presented in Results chapter, but their reorganization in Material and Methods may be considered.

Response: Thanks for your suggestions. In the Materials and Methods part, we have added some details to the description of the experimental method.

An interpretation of „a” and „b” indexes used in Table 6 is better to be given at the bottom of this table.

Response: Thanks for your advice. We have added a note in Table 6. As follows:

Values with different superscripts within the same line differ significantly at P<0.01 (A, B, C) and P<0.05 (a, b, c).

Thanks again for your selfless contribution on the manuscript.

Round 2

Reviewer 2 Report

No comments.

Author Response

Thanks a lot to the reviewer.

Reviewer 4 Report

Undoubtedly, the new version subscribed by the authors is more complete and at a higher level from a scientific point of view. Improvements were found in most work points, including the statistics addressed, the results presented and their discussion.

Author Response

Thanks to the reviewer for these motivating comments.